# Effect of Copper Chelators via the TGF-β Signaling Pathway on Glioblastoma Cell Invasion

**DOI:** 10.3390/molecules27248851

**Published:** 2022-12-13

**Authors:** Heabin Kim, Seonmi Jo, In-Gyu Kim, Rae-Kwon Kim, Yeon-Jee Kahm, Seung-Hyun Jung, Jei Ha Lee

**Affiliations:** 1Department of Genetic Resources, National Marine Biodiversity Institute of Korea, Seocheon 33662, Republic of Korea; 2Department of Radiation Biology, Environmental Safety Assessment Research Division, Korea Atomic Energy Research Institute, Daejeon 34057, Republic of Korea; 3Department of Radiation Science and Technology, Korea University of Science and Technology, Daejeon 34113, Republic of Korea

**Keywords:** copper, D-penicillamine, triethylenetetramine, EMT, glioblastoma, zebrafish

## Abstract

Glioblastoma multiforme (GBM) is a fast-growing and aggressive type of brain cancer. Unlike normal brain cells, GBM cells exhibit epithelial–mesenchymal transition (EMT), which is a crucial biological process in embryonic development and cell metastasis, and are highly invasive. Copper reportedly plays a critical role in the progression of a variety of cancers, including brain, breast, and lung cancers. However, excessive copper is toxic to cells. D-penicillamine (DPA) and triethylenetetramine (TETA) are well-known copper chelators and are the mainstay of treatment for copper-associated diseases. Following treatment with copper sulfate and DPA, GBM cells showed inhibition of proliferation and suppression of EMT properties, including reduced expression levels of N-cadherin, E-cadherin, and Zeb, which are cell markers associated with EMT. In contrast, treatment with copper sulfate and TETA yielded the opposite effects in GBM. Genes, including *TGF-β*, are associated with an increase in copper levels, implying their role in EMT. To analyze the invasion and spread of GBM, we used zebrafish embryos xenografted with the GBM cell line U87. The invasion of GBM cells into zebrafish embryos was markedly inhibited by copper treatment with DPA. Our findings suggest that treatment with copper and DPA inhibits proliferation and EMT through a mechanism involving TGF-β/Smad signaling in GBM. Therefore, DPA, but not TETA, could be used as adjuvant therapy for GBM with high copper concentrations.

## 1. Introduction

Glioblastoma multiforme (GBM) is one of the most malignant and aggressive types of brain tumor in adults, with a five-year survival rate of approximately 5%. Unlike normal brain cells, GBM cells exhibit epithelial–mesenchymal transition (EMT), which occurs during embryonic development or tissue remodeling in many species and is involved in cancer cell metastasis [1,2,3]. EMT is associated with migratory and invasive properties in tumor progression, which can lead to the spread of the tumor to other parts of the body through the blood and lymph nodes. Given the role of EMT in the onset of tumor metastasis, modulating the EMT is currently considered a promising strategy for inhibiting cancer metastasis and improving patient survival [4].

Transforming growth factor-β (TGF-β) is a cytokine involved in cell growth, inflammatory processes, tumorigenesis, and cell differentiation [5]. TGF-β has been reported to function as a protumorigenic factor by promoting cancer cell dissemination and metastasis, as well as being involved in glioma invasion, high-grade tumors, and poor clinical prognosis, which is associated with TGF-β expression levels [6,7,8,9]. TGF-β can be divided into three types: TGF-β1, TGF-β2, and TGF-β3. Analysis of the structures of the isoform revealed approximately 76% amino acid sequence homology in various tissues [10]. TGF-β is known to contribute to EMT regulators, such as Snai1 and Zeb1, which are involved in a variety of processes ranging from tumorigenesis to interaction with transcription factors [6,11].

Copper, a trace mineral, is an essential component for the homeostasis of physiological functions such as metabolism and nutrient absorption. The imbalance of Cu in humans causes serious disorders such as Wilson’s disease and Menke’s disease. Accumulating scientific evidence shows that Cu plays a critical role in the proliferation, angiogenesis, and metastasis of a variety of cancers [12,13,14]. Excess copper with unpaired electrons, such as hydroxyl and superoxide free radicals, can favor lipid peroxidation, DNA strand breakage, and protein damage [15,16]. Thus, copper levels in biological systems are strictly controlled through the actions of copper transporters and related proteins [17,18]. Recent studies have shown that aberrantly increased levels of copper in human malignancies and its depletion suppress many cell signaling pathways, leading to inhibition of tumor growth and reduced metastasis [19]. In GBM, excessive copper levels were found in the tumor regions compared with the levels in normal tissues in humans [20]. We used FDA-approved copper chelators to introduce suitable copper ions into cells to regulate malignant processes and limit cancer metastasis. D-penicillamine (DPA) is a well-known degradation product of penicillin and is mainly used as a copper-chelating agent. DPA binds to excess copper through sulfhydryl and amino groups to form a ring complex and mobilizes intracellular copper into circulation. However, DPA causes various adverse effects, such as headaches, bone marrow suppression, skin toxicity, and immune disorders [21]. Therefore, DPA has been replaced with alternative chelating agents, such as trientine (TETA). TETA is a selective chelator that aids in the elimination of copper from the human body by forming a complex that is excreted from the kidney. Unlike DPA, TETA was introduced as a better therapeutic option for patients with Wilson’s disease and was found to be potentially free of the side effects caused by DPA [22].

Recently, zebrafish have been increasingly used in cancer research [23,24,25]. Zebrafish xenograft models are widely used to evaluate cancer progression, neoangiogenesis, and drug response/resistance due to the high degree of physiological and genetic similarity with the mammalian tumor microenvironment [26]. In zebrafish, cancer cells can be fluorescently labeled and directly observed in living animals owing to the optical clarity of zebrafish larvae.

In this study, we determined the role of copper chelation in regulating the remodeling of the cytoskeleton and its association with EMT-related transcription factors in GBM using a model of zebrafish xenografted with GBM cells.

## 2. Results

### 2.1. TGF-β Expressions and Morphological Changes with CuSO_4_ in GBM Cells

To confirm the effect of CuSO_4_ on GBM cells, cellular shape and movement were observed. Morphological changes, which are a key characteristic of the EMT, from a spindle shape to a cobblestone-like shape were observed in cells treated with 100 μM CuSO_4_ when compared to the untreated control cells (Figure 1A). Previous studies have reported a 70 to 100 μM range of physiological Cu concentrations in cerebrospinal fluid and the synaptic cleft [27,28]. TGF-β is a potent EMT regulator that leads to enhanced migration and infiltration of GBM cancers, which is a more common feature of mesenchymal cells [10]. To confirm changes at the cellular level of TGF-β, we treated cells with CuSO_4_ (25, 50, and 100 µM) for 24 h. The expression levels of *TGF-β* decreased after treatment with 100 µM CuSO_4_, determined using Western blot (Figure 1B). In addition, the transcriptional levels of *TGF-β1* and *TGF-β2* were inhibited in U87 and U251 cells treated with 100 µM CuSO_4_ (Figure 1C). These results suggest that the interaction between copper and the TGF-β isotype may be associated with EMT.

### 2.2. Regulation of Cellular Viability by CuSO_4_ and Chelators

To regulate the effect of CuSO_4_, we used drugs with suitable copper ions in GBM cells and limited cancer metastasis. We confirmed the relative cytotoxicity of different concentrations of copper and chelators in U87 and U251 cells. Our results indicate that the IC50 of copper sulfate is approximately 50 μM, similar to the reported IC50 value in glioblastoma cells [29]. In addition, no toxic effect was observed at concentrations of up to 200 μM on chelators (TETA or DPA) in GBM cells (Appendix A). The cytotoxicity of CuSO_4_ was tested at three concentrations (25, 50, and 100 μM) with 100 μM chelators using a CCK-8 assay in GBM cells. As shown in Figure 2A, we observed a dose-dependent decrease in viability in cells treated with only CuSO_4_, whereas the cell viability of up to 65% of the cells treated with the Cu/TETA complex was restored. On the other hand, GBM cells treated with the Cu/DPA complex had a lower survival percentage than those treated with only DPA and CuSO_4_. Treatment with various concentrations of chelators in 50 μM of CuSO_4_ increased the cell viability of cells treated with Cu/TETA but decreased the cell viability of those treated with the Cu/DPA complex (Figure 2B). The cytotoxicity of DPA and TETA was significantly stronger than that of the Cu or Cu/DPA complex at all concentrations in both cell lines. We further investigated the combined effect of CuSO_4_ and chelators on tumor growth using a colony formation assay. The colony formation assay showed that cells treated with Cu/TETA had significantly restored cellular growth compared with cells treated with CuSO_4_ alone. In contrast, treatment with DPA inhibited cancer cell growth (Figure 2C), and cells treated with CuSO_4_ and DPA changed from a spindle shape to a cobblestone-like shape (Figure 2D). This result indicates that the treatment of GBM cells with the Cu/DPA complex may affect cell cytotoxicity more than treatment with CuSO_4_ or Cu/TETA.

### 2.3. Effect of CuSO_4_ and Chelators on EMT

EMT is involved in wound repair and aggressive phenotypes in a variety of tissues. The wound healing assay indicated a remarkable increase in the motility of copper-exposed U87 and U251 cells after 48 h. GBM cells treated with Cu/TETA showed significantly increased wound healing capacity compared with those treated with CuSO_4_, whereas Cu/DPA inhibited the wound healing capacity. Thus, regulating CuSO_4_ using chelators involved in cell motility reduced EMT progression (Figure 3A). We evaluated the migration/invasion assay and EMT regulatory proteins in copper-exposed cells using RT-PCR. The migration/invasion assay indicated that the combined treatment of CuSO_4_ and DPA was highly effective in inhibiting tumor cell migration and invasion in both cell lines (Figure 3B). This is consistent with the expression of EMT regulators, such as N-cadherin, E-cadherin, and vimentin, as well as decreased levels of transcription factors Zeb1 and Snail in GBM cells using RT-PCR and immunofluorescence (Figure 3C,D). Our experimental results showed that CuSO_4_ suppresses the invasive and migratory capacities of GBM cells, and this effect was synergistically increased by combinatorial treatment with Cu/DPA. In contrast, treatment with CuSO_4_ and TETA yielded opposite results.

### 2.4. Correlations between TGF-β and EMT Markers in GBM Cells

Considering that the TGF-β pathway occurs through phosphorylation of Smad2/Smad3, we investigated the cellular levels of TGF-β isotypes and signaling molecules. As a result, GBM cells treated with Cu/DPA showed decreased transcriptional levels of *TGF-β1* and *TGF-β2* compared with cells treated with CuSO_4_ or Cu/TETA, as determined by RT-PCR (Figure 4A). Consistent with our Western blotting assay, the effect of copper with chelators on TGF-β signaling in GBM cells was demonstrated by the decreased expression levels of *TGF-β* and decreased phosphorylation of Smad2/Smad3 (Figure 4B). To confirm the role of TGF-β/Smad signaling in GBM cells, we used CuSO_4_ and TGF-β to regulate wound healing and the levels of EMT markers, such as N-cadherin, E-cadherin, and vimentin. Wound healing assays were also used to investigate the absence or presence of 40 ng/mL TGF-β1 in the presence of CuSO_4_, and images were obtained at 48 h (Figure 4C). Consistently, we evaluated the profile of EMT markers in relation to treatment with copper and TGF-β by Western blotting assays (Figure 4D). These results suggest that the association between copper (with chelators) and TGF-β may be involved in EMT-associated properties. To directly confirm the link between TETA or DPA-mediated copper depletion, we used a copper assay kit to measure the changes in copper concentrations, which may be related to increased invasiveness and metastatic potential. Interestingly, only TETA decreased the copper concentration in a dose-dependent manner compared to DPA (Figure 4E). Therefore, we suggest that the Cu/DPA-induced antimetastatic effect contributes to the inhibition of TGF-β signaling and downregulation of EMT in GBM cells.

### 2.5. Invasion Capability of U87 Cells with CuSO_4_ and/or Chelators in Zebrafish Embryos

To show the invasiveness of U87 cells in the tail region of the embryos via the vessels, we used an uplight digital fluorescence stereomicroscope to obtain high-resolution images. CM-DiI-positive cells in the tail were quantified to determine invasion. The cells migrated to the edge of the embryonic yolk sac, distant from the original site of injection. At concentrations above 4 µM CuSO_4_, the survival rate was significantly reduced in zebrafish. The group treated with CuSO_4_ and chelators had a significantly restored viability compared to the group treated with CuSO_4_ alone (Figure 5A). However, treatment with CuSO_4_ and DPA suppressed cancer cell mobility and invasiveness in zebrafish models (Figure 5B). These results confirmed that the zebrafish xenograft model can be used for functional studies on the role of CuSO_4_ and/or chelators in U87 cells.

## 3. Discussion

Discovering new therapeutic uses for existing drugs is a valuable approach as their safety profile is well known. The observation that excess copper was required for proliferation and invasion in cancer led to the initiation of clinical trials that evaluated the impact of copper chelation in patients with GBM. Clinically, DPA and TETA have been widely used and extensive preclinical and clinical data are available [22,30,31]. DPA is known as a drug for penicillin hydrolysis and is used to treat Wilson’s disease and cancer. Copper chelation with DPA, in combination with a low-copper diet and 6000 cGy radiation, has been evaluated in clinical trials involving patients with GBM. Unfortunately, no significant impact on the clinical outcomes of GBM has been observed using this therapeutic strategy [32]. The use of TETA in combination with carboplatin has been reviewed in patients with platinum-resistant ovarian cancer, and severe adverse effects were observed in these patients [33]. Copper metabolism has emerged as an attractive target for therapy development, and several drugs, including DPA, TEPA, and other inhibitors, have been used in various cancer trials. However, drugs targeting altered metabolism during metastasis have not yet been elucidated. Our findings identify copper chelators involved in the metastasis axis of TGF-β and highlight its potential to be developed as a next-generation therapeutic approach for high-risk patients with GBM. The gene expression profile revealed that *TGF-β* was significantly upregulated in GBM cells and that *TGF-β* expression correlated with clinicopathological parameters (Appendix A). In addition, Kaplan–Meier survival analysis revealed an association of TGF-β in GBM and lower grade glioma (LGG) using a publicly accessible database, the gene expression database (TIMER; https://cistrome.shinyapps.io/timer/, accessed on 2 November 2022). Therefore, we suggest that stimulation of excess CuSO_4_ with a chelator in GBM is an important cause of the downregulation of *TGF-β* expression. This study provides support for the use of copper with chelator complexes as potential strategies for cancer therapy.

TGF-β is a multifunctional growth factor that plays a key role in various cellular actions such as cell proliferation and metastasis. In a recent study, glioma with high *TGF-β* expression was found to recruit CD133, which is involved in stem cells. Furthermore, neutralization of TGF-β in glioma stem cells inhibits their invasiveness [34]. There is an established signaling pathway that acts through TGF-βR1/2 heterodimer receptors to phosphorylate Smad2 and Smad3, which are subsequently imported into the nucleus to regulate the expression of specific genes. Pathological forms of TGF-β signaling created by cancer cells can lead to EMT and ultimately cause invasiveness. The level of *TGF-β* expression was increased in chondrocytes of newborn pigs following treatment with 32 µM CuSO_4_ at 48 h [35] and in mice following treatment with 40 mg/kg CuSO_4_ on day 48 [36]. In contrast, the expression of *TGF-β1* and *IGF1* is inhibited by up to 150 µM copper chloride in osteoblasts [37]. These findings suggest that excess CuSO_4_ may regulate TGF-β-related EMT progression in an in vitro environment. It is widely believed that TGF-β signaling has evolved to enforce premalignant cells, resulting in tumorigenic effects in cancer. In this study, the suppression of TGF-β decreased the phosphorylation of Smad2/3 in GBM cells treated with CuSO_4_ and DPA. Zeb1 and Snail are transcription factors that induce EMT and are characterized by an increase in cell spreading and cell–cell separation via the regulation of E-cadherin. The induction of Snail may be a Smad3-dependent process. TGF-β induces the Smad-independent pathway, and RAS/MAPK is vital to EMT and plays a role in the cellular expression of genes and metastasis [36]. Specific factors play important roles as EMT regulators and transcription factors. EMT regulators are structural proteins of cells such as N-cadherin, E-cadherin, and vimentin. Snail and Zeb1 are transcription factors, and their overexpression is involved in the invasiveness, metastasis, and poor prognosis of GBM. Upregulation of Snail is associated with the promotion of EMT, primarily through inhibition of E-cadherin [3,7]. The Cu/DPA complex reversed the elongated, fibroblast-like shape of the cells back to the epithelial shape. From the study of cell function and protein expression, we observed that Cu/DPA suppressed the expression of TGF-β and EMT markers in GBM cells, but we still need to determine the precise mechanism by which the copper chelator inhibits this pathway. This demonstrates that Cu or DPA treatment lead to the inhibition of the TGF-β/Smad pathway which subsequently suppress EMT in GBM cells. On the other hand, TETA can restore copper-reduced EMT by increasing the expression of mesenchymal proteins, which induce cell migration and invasion. We investigated the ability of TGF-β to recover CuSO_4_-reduced EMT in GBM cells. These findings suggest an approach for the regulation of CuSO_4_-suppressed EMT by TGF-β treatment in GBM cells. Further investigations focused on metastatic zebrafish models to determine how DPA inhibits the metastasis of GBM. We showed that CuSO_4_ or chelators at concentrations as high as 100 µM did not affect cell viability in vitro. A previous study showed that 10 μM Cu did not induce mortality [38]. However, because of the high mortality induced by exposure to CuSO_4_ (at concentrations higher than 5 µM) in the zebrafish model test, we treated the cells with 4 µM copper and 10 µM chelators.

The results showed that only CuSO_4_ and Cu/DPA significantly reduced cell invasion and metastatic dissemination in zebrafish, supporting the role of DPA as a potent antimetastatic agent. To directly establish a link between TETA- or DPA-mediated copper depletion and metastasis in GBM cells, we used a copper assay kit to confirm the changes in copper concentration, which is related to increased invasiveness and metastatic potential (Figure 5). Our results are in agreement with a recent study showing that the Cu/DPA complex can induce apoptosis in U251 cells [29]. We confirmed the induction of reactive oxygen species and apoptosis using the expression of apoptosis markers, such as Bax and Bcl-2, in Cu/chelator-treated U87 and U251 cells (Appendix A). Although several studies and clinical trials have shown that Cu and chelators are effective in inhibiting cancer proliferation and angiogenesis, we showed that copper chelators respond to different effects of cytotoxicity and that DPA could be repurposed as a metastatic inhibitor in excess-Cu environments. We have provided crucial evidence for using the Cu chelators TETA and DPA to respond to cancer migration/invasion properties and improve the survival of GBM cells and injected zebrafish models. Although the increased survival of zebrafish in the Cu/DPA group may be a consequence of a decreased concentration of Cu or water, this is an impressive result considering the metastatic nature of this model. Neither DPA or TETA affected tumor cell growth or EMT properties in GBM cells and xenograft models. We demonstrated that DPA suppresses the expression of TGF-β signaling properties in the presence of copper and exhibits concentration-dependent cytotoxicity in GBM cells. The recovery of cell survival and metastasis observed in response to Cu/TETA could be a consequence of the reduction in the Cu concentration, which was determined using the copper assay kit. Conversely, the use of DPA did not affect the level of Cu but caused a decrease in the expression of TGF-β and EMT in GBM cells.

## 4. Materials and Methods

### 4.1. Cell Culture

U87 and U251 cell lines used in the study were obtained from the American Type Culture Collection (ATCC, Manassas, VA, USA) and were grown in DMEM medium (Invitrogen, Carlsbad, CA, USA) supplemented with 10% fetal bovine serum (HyClone, Logan, UT, USA) and penicillin/streptomycin (HyClone). Cells were incubated at 37 °C in a humidified atmosphere with 95% air/5% CO_2_.

### 4.2. Reverse Transcription-PCR (RT-PCR)

Total RNA was isolated from cells under various conditions with TRIzol reagent (Invitrogen). First-strand cDNA was generated from 1 µg of total RNA using oligo-dT primers and a cDNA synthesis kit (iNtRON Biotechnology, Gyungki-do, Korea). Resultant cDNA served as templates for PCR amplification with specific primers. PCR conditions were as follows: initial denaturation at 94 °C for 5 min, 35 cycles of 94 °C for 1 min, 56 °C for 1 min, and 72 °C for 90 s, and a final extension at 72 °C for 10 min. The amplified PCR products were analyzed on 1% agarose gels (Bio Basic) using ethidium bromide. TGF-β and EMT-related primer sequences for RT-PCR are presented in Appendix A.

### 4.3. Reagents and Antibodies

TGF-β (T7039) and copper sulfate (C8027) were obtained from Sigma-Aldrich (St. Louis, MO, USA). D-penicillamine (Sigma) and trientine hydrochloride (Sigma) were used as chelators of copper. Antibodies specific for β-actin (47778), Vimentin (6260), N-cadherin (59987), and E-cadherin (8426) were purchased from Santa Cruz Biotechnology. Antibodies for TGF-β (3711), Smad2/3 (3102), p-Smad2 (18338), and p-Smad3 (9520) were purchased from Cell Signaling Technology (Beverly, MA, USA).

### 4.4. Western Blot Analysis

Glioblastoma cells were lysed in lysis buffer with protease inhibitor cocktails (Sigma-Aldrich, St. Louis, MO, USA). Protein concentrations were determined by Bradford assay (Bio-Rad, Hercules, CA, USA). For Western blot analysis, equal amounts of protein were separated on 8–15% sodium dodecyl sulphate (SDS)-polyacrylamide gels and transferred to PVDF membranes. Blots were blocked for 1 h at room temperature with blocking buffer. Membranes were incubated overnight in a cold chamber with specific antibodies. After being washed with Tris-buffered saline (TBS), membranes were incubated with a horseradish-peroxidase-labeled secondary antibody (Abcam) and visualized using a Supersignal west atto ultimate sensitivity substrate (Thermo Scientific, Waltham, MA, USA; A38555).

### 4.5. Wound Healing Assay

The GBM cells were seeded at 1 × 10^6^ cells in a 6-well plate to full confluence and then serum-free overnight. Cells were scratched with a pipette tip across the center of the wells. Images were captured using a light microscope 24–48 h after treatment at 37 °C with the Cu/chelators. The healing area was quantified by measuring the distances between 10 randomly selected points within the wound edge, and the mean values and standard deviations were plotted.

### 4.6. Invasion and Migration Assay

Cell invasion and migration assays were performed using matrigel-coated invasion/noncoated migration chambers (8 μm pores; BD Biosciences, San Jose, CA, USA). Cells were seeded in the upper chamber at 2 × 10^5^ cells in 200 µL of serum-free medium/well and incubated for 24 h at 37 °C in a humidified atmosphere of 5% CO_2_. Nonmigratory cells in the upper chamber were removed by wiping with a cotton swab. The stained cells were counted under a light microscope in four randomly chosen fields, with results expressed as means of five cases from a representative of two independent experiments.

### 4.7. Colony Forming Assay

Cells were plated in 35 mm culture dishes at a density of 1 × 10^3^ cells per plate and allowed to attach overnight. Cells were left untreated or exposed to a 2 Gy dose of radiation. Cells were incubated for 10–14 days and stained with 0.5% crystal violet. Colonies, defined as groups of ≥50 cells, were counted. Clonogenic survival was expressed as a percentage relative to the untreated controls.

### 4.8. Fluorescence Microscopy

Cells (5 × 10^4^) were cultured on glass coverslips in six-well plates and fixed with 4% paraformaldehyde. Following cell fixation, cells were incubated with anti-N-cadherin antibody in a solution of PBS with 1% bovine serum albumin and 0.1% Triton X-100 at 4 °C overnight. Staining was visualized using Alexa Flour 488–conjugated antirabbit IgG antibody (Invitrogen). Nuclei were counterstained using 4,6-diamidino-2-phenylindole (Sigma-Aldrich). Stained cells were analyzed using a Zeiss LSM510 Meta microscope (Carl Zeiss MicroImaging GmbH, Göttingen, Germany).

### 4.9. Copper Assay Kit

The release of copper from treatment with Cu/chelators was evaluated in PBS using a copper assay kit (Sigma, MAK127). PBS with Cu/chelators was incubated for 10 min with a colorimetric reagent and analyzed at 359 nm with the spectrophotometer. A standard curve was created by plotting the absorbance values of the 100, 200, and 300 µg/dL standards against the concentration of copper in these standards (R2 = 0.99), and a regression equation was used to calculate the amount of copper (μM) in the samples. This assay can detect copper concentrations up to 47 μM.

### 4.10. Zebrafish Xenograft

U87 cells were stained with 2 μM CM-DiI (Invitrogen, Carlsbad, CA, USA) dye for 10 min at 37 °C, following by 15 min at 4 °C. For microinjection of cells into the embryos, zebrafish embryos of 2 days post fertilization (dpf) were anesthetized in tricaine, positioned on a 1.2% low-melting agarose gel, and injected with 500 cells into the middle of the embryonic yolk sac region using a Pneumatic Pico-Pump Injector. After injection, the xenografts were cultured at 34 °C and the zebrafish larvae with similar sizes of transplanted cells were isolated in a 24-well plate to treat with CuSO_4_ and chelators. At 5 days post injection, the zebrafish were mounted on 3% methylcellulose (Sigma). Images were made with a Leica DM6 B fluorescent microscope (Leica, Wetzlar, Germany).

## 5. Conclusions

We analyzed the morphology and mechanism of TETA and DPA against the CuSO4-induced inhibitory effect on EMT in zebrafish models using migration/invasion-based TGF-β signaling. Alterations in the identified gene and protein biomarkers might help elucidate the effect of TETA and DPA against the CuSO4-induced inhibitory effect of EMT. Our results also provide substantial evidence of a potential novel and sequential mechanism of the inhibition of EMT induced by CuSO4, as well as a multicomponent and multiregulatory therapeutic mechanism for copper chelators.

## Figures and Tables

**Figure 1 molecules-27-08851-f001:**
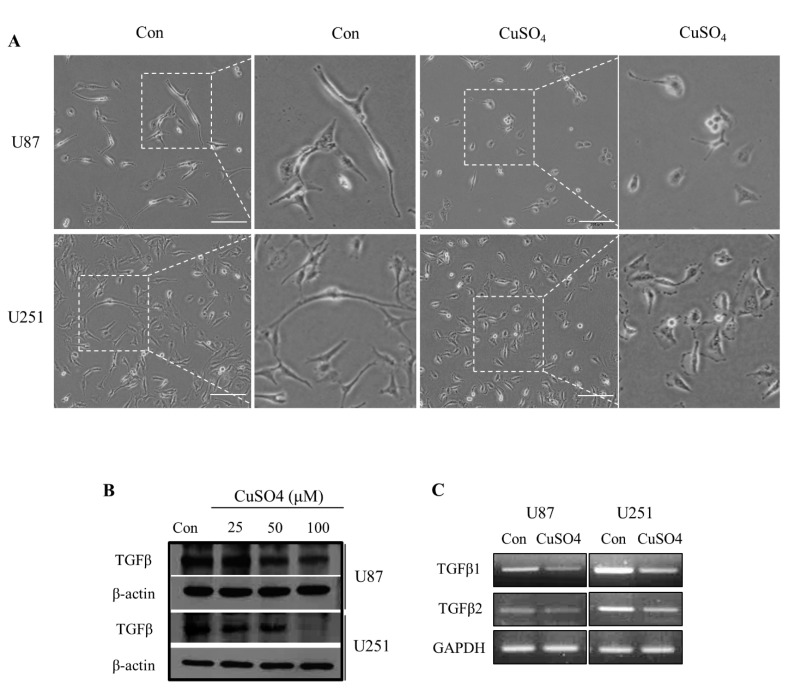
(**A**) The effect of control and 100 μM CuSO_4_ on morphology of U87 and U251 cells seeded in dishes and treated for 24 h. Scale bars: 100 µm. (**B**) The change in expression levels of TGF-β in GBM cells treated with CuSO_4_ (25, 50, and 100 µM). (**C**) The mRNA expression levels of TGF-β1 and 2 in cells treated with 100 µM CuSO_4_.

**Figure 2 molecules-27-08851-f002:**
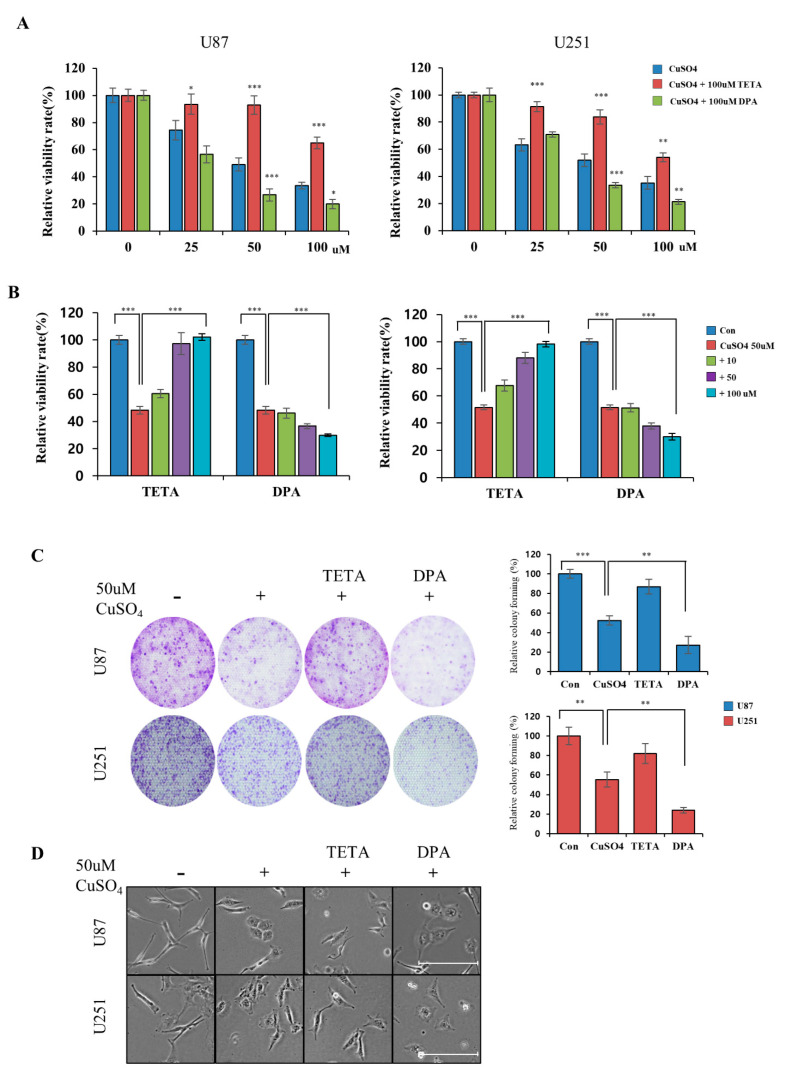
(**A**) The cytotoxicity at concentrations of 25, 50, and 100 μM CuSO_4_ with 100 μM chelators (TETA or DPA) using a CCK-8 assay in GBM cells. (**B**) Cellular viability using CCK-8 assay in GBM cells treated with various concentrations of chelators in 50 μM of CuSO_4_. (**C**) Colony formation in GBM cells treated with 50 μM of CuSO_4_ and 100 μM chelators. (**D**) Morphology of U87 and U251 cells treated with 50 μM of CuSO_4_ and 100 μM chelators. Scale bars: 100 µm. Data represent mean ± SD of three independent experiments using two-tailed *t*-test. * *p* < 0.05, ** *p* < 0.01, *** *p* < 0.001.

**Figure 3 molecules-27-08851-f003:**
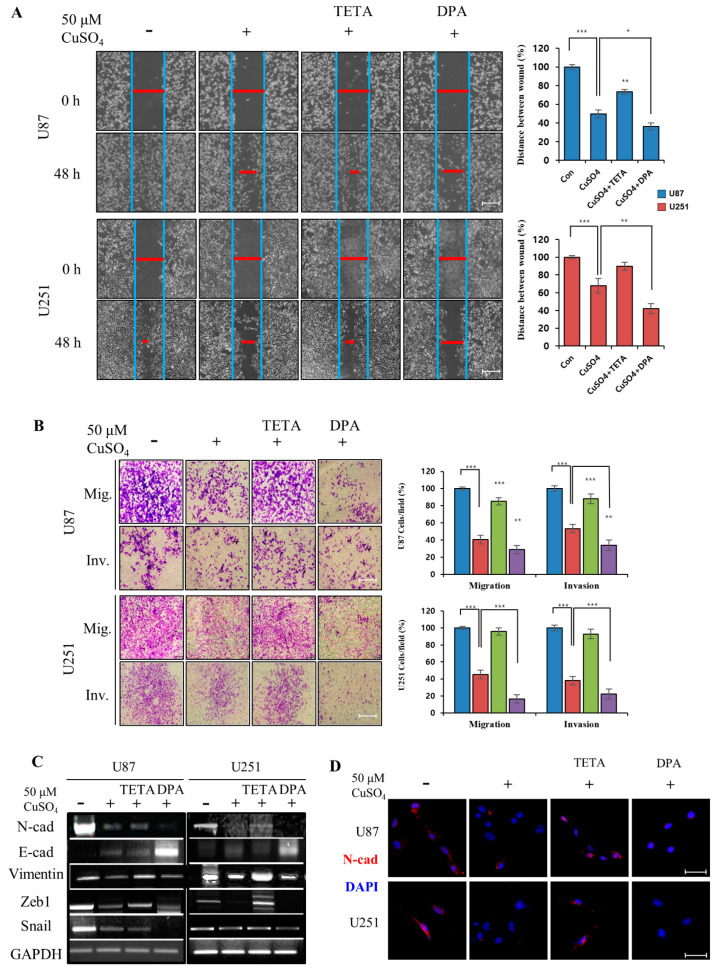
(**A**) Wound healing capacity of U87 and U251 cells treated with 50 μM of CuSO_4_ and 100 μM chelators. Scale bars: 200 µm. (**B**) The migration/invasion assay indicating that treatment of CuSO_4_ and DPA was highly effective in GBM cell migration and invasion in both cell lines. Scale bars: 200 µm. (**C**) The expression of EMT regulators and transcription factors in GBM cells using RT-PCR. (**D**) Immunofluorescent analysis of N-cadherin, after treatment with CuSO_4_ with chelators in GBM cells. Scale bars: 100 µm. Data represent mean ± SD of three independent experiments using two-tailed *t*-test. * *p* < 0.05, ** *p* < 0.01, *** *p* < 0.001.

**Figure 4 molecules-27-08851-f004:**
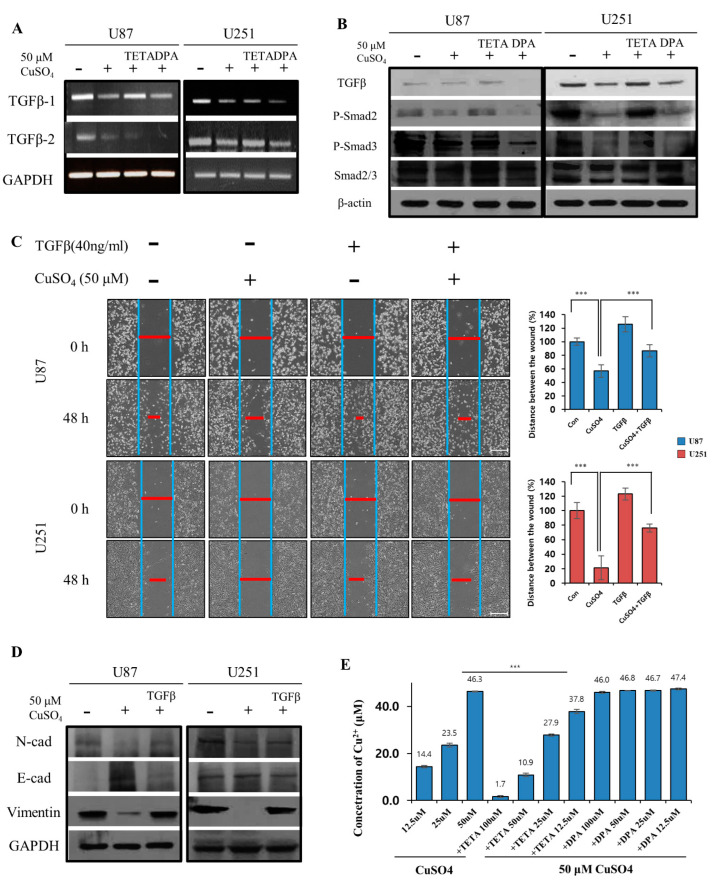
(**A**) The transcriptional levels of TGF-β1 and TGF-β2 compared with cells treated with CuSO_4_ and chelators using RT-PCR. (**B**) The effect of copper with chelators on TGF-β/Smad signaling in GBM cells evaluated by Western blotting assay. (**C**) Wound healing assays of absence or presence of 40 ng/mL TGF-β1 in 50 μM CuSO_4_ and images obtained at 48 h. Scale bars: 200 µm. (**D**) The expression of EMT regulators in GBM cells evaluated using Western blot assay. (**E**) Measurement of a copper concentration treated with 50 μM CuSO_4_ with various concentrations of chelators using a copper assay kit. Data represent mean ± SD of three independent experiments using two-tailed *t*-test. *** *p* < 0.001.

**Figure 5 molecules-27-08851-f005:**
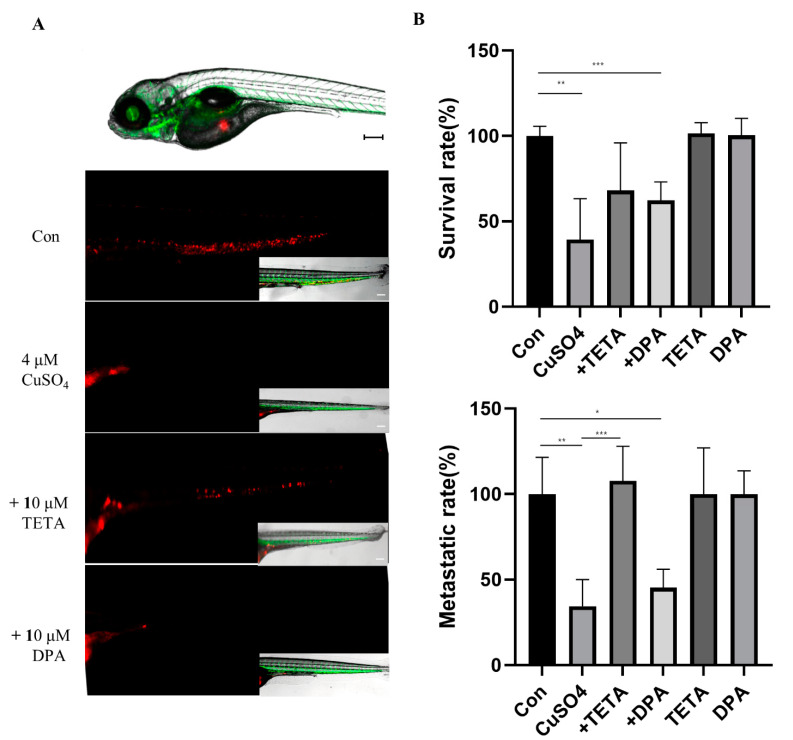
(**A**) Representative images at higher magnification show the invasive U87 cells (red) in the tail region of the embryos via vessels (green). Scale bars: 200 µm. (**B**) At 5 days post injection, the survival and metastatic rate of Tg (kdrl:EGFP) zebrafish embryos microinjected with CM-DiI U87 cells (n = 50/each group). * *p* < 0.05, ** *p* < 0.01, *** *p* < 0.001.

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
