# Peer review of "Effect of Copper Chelators via the TGF-β Signaling Pathway on Glioblastoma Cell Invasion"

_molecules, 2022, doi:10.3390/molecules27248851_

Round 1

Reviewer 1 Report

Line 24: Gliobalstoma Multiforme is not the most common brain cancers. Glioblastoma is common not GBM.

Copper has not been discussed anywhere in materials and methods.

Control is missing for cell line experiments

Why was RT-PCR performed before the treatment.

Table 1 is missing

Positive control (any reference drug) not included in study ??

IC50 not calculated for copper 

On what basis  50μM was selected.

Figure 2 

why green bar shows more than 100% viability at 0 concentration ?? U87

why bars shows variation in viability at 0 concentration ?? U251

Scale bars missing for these figures  3B, 3D, 5A

Figure 5 positive control missing ???

2D (scale bar not proper), 

Author Response

à Please see corrections made in response to reviewer comments in blue in the main text.

Reviewer 1

→ We thank the reviewer for their valuable comments. We have corrected the errors pointed out by the reviewer and checked the entire manuscript again.

Q1. Line 24: Gliobalstoma Multiforme is not the most common brain cancers. Glioblastoma is common not GBM.

à We corrected to “a fast-growing and aggressive type of brain cancer.” (Line 24)

Q2. Copper has not been discussed anywhere in materials and methods.

à We added information about copper sulfate. (Line 112)

Q3. Control is missing for cell line experiments

à We added Supplementary Fig1.

Q4. Why was RT-PCR performed before the treatment.

à We observed the change of cellular structure dependent on the concentration of CuSO4. We were interested in TGFβ isomers associated with poor survival in patients with GBM cancer and therefore conducted RT-PCR.

Q5. Table 1 is missing à We added Supplementary Table 1.

Q6. Positive control (any reference drug) not included in study ??à We added Supplementary Fig1.

Q7. IC50 not calculated for copper on what basis 50μM was selected.

à We modified the sentence in the manuscript (Line 197)

Q8. Figure 2

why green bar shows more than 100% viability at 0 concentration ?? U87

why bars shows variation in viability at 0 concentration ?? U251

à Thank you for pointing it out. We have corrected it.

Q9. Scale bars missing for these figures 3B, 3D, 5A

à Thank you for pointing this out. We have added scale bars.

Q10. Figure 5 positive control missing ???

à We have corrected Fig5.B.

Q11. 2D (scale bar not proper)

à We have corrected the scale bars.

Reviewer 2 Report

In this study, Kim and colleagues investigate the effect of copper chelators on glioblastoma (GBM) cells. The authors first examined the effects of CuSO4 on GBM cells and found that CuSO4 inhibits the EMT phenotypes and downregulates TGF-b expression in GBM cells. The authors then evaluate the effects of two copper chelators, TETA and DPA, on GBM cells and showed those two copper chelators have distinct effects on GBM cells growth. Consistent with these differences, the authors also observed different EMT phenotype upon treatment with TETA and DPA. Lastly, the authors showed that DPA treatment inhibits the invasion of GBM cells in zebrafish embryos and the mechanism is likely through inhibition of TGF-b signaling. Overall, the data clearly support the different effects of TETA and DPA on GBM cells growth and invasion. However, there are several issues need to be addressed to better support the conclusions of the current study.

1. The whole study is based on assumption that cooper levels affect GBM cells growth and metastasis. To examine the effect of cooper chelators, the authors treated cells with 100uM CuSO4, which could be way higher than any physiological condition. The authors should justify the relevance of their in vitro experimental condition.

2. It appears the GBM cells used in this study are already in mesenchymal state, therefore it is necessary to clarify the effects of CuSO4 is through MET.

3. The authors examined TGF-b expression by western blot, which only detect the precursor of active TGF-b. Therefore, the authors need to provide evidence to show that these TGF-b have activity in vitro.

4. TGF-b signaling can regulates tumor growth through inhibition of cell proliferation or induce apoptosis. The authors should distinguish the effects on those two aspects in their assays.

5. It is important to show if DPA alone has any effects on EMT phenotypes in GBM cells.

6. It is known that other cells in the tumor microenvironment can be important source of TGF-b rather than cancer cell themselves. Therefore, the effect of DPA in vivo would be more complicated.

Author Response

à Please see the corrections made in response to reviewer comments in blue in the main text.

Reviewer 2.

In this study, Kim and colleagues investigate the effect of copper chelators on glioblastoma (GBM) cells. The authors first examined the effects of CuSO4 on GBM cells and found that CuSO4 inhibits the EMT phenotypes and downregulates TGF-b expression in GBM cells. The authors then evaluate the effects of two copper chelators, TETA and DPA, on GBM cells and showed those two copper chelators have distinct effects on GBM cells growth. Consistent with these differences, the authors also observed different EMT phenotype upon treatment with TETA and DPA. Lastly, the authors showed that DPA treatment inhibits the invasion of GBM cells in zebrafish embryos and the mechanism is likely through inhibition of TGF-b signaling. Overall, the data clearly support the different effects of TETA and DPA on GBM cells growth and invasion. However, there are several issues need to be addressed to better support the conclusions of the current study.

à We thank the reviewer for their valuable comments and suggestions made to enhance the value of this research.

Q1. The whole study is based on assumption that cooper levels affect GBM cells growth and metastasis. To examine the effect of cooper chelators, the authors treated cells with 100uM CuSO4, which could be way higher than any physiological condition. The authors should justify the relevance of their in vitro experimental condition.

à In the revised manuscript, we modified the sentence: "Previous studies have reported a 70 and 100 uM range of physiological Cu concentrations in cerebrospinal fluid and the synaptic cleft." (line 181)

Q2. It appears the GBM cells used in this study are already in mesenchymal state, therefore it is necessary to clarify the effects of CuSO4 is through MET.

à We do agree with the reviewer’s point about mesenchymal to epithelial transition (MET) of GBM cells as mediated by CuSO4. We understand that reduced EMT is consistent with promoting MET (thereby suppressing EMT).

Q3. The authors examined TGF-b expression by western blot, which only detect the precursor of active TGF-b. Therefore, the authors need to provide evidence to show that these TGF-b have activity in vitro.

àTo examine the EMT involved in the activation of TGFβ, we confirmed the phosphorylation of Smad2/Smad3 in Fig. 4B. There is an established signaling pathway that acts through TGFβ receptors to phosphorylate Smad2 and Smad3, which are subsequently imported into the nucleus. Pathological forms of TGF-β signaling created by cancer cells can lead to EMT and ultimately cause invasiveness. Suppression of TGFβ decreased the phosphorylation of Smad2/3 in GBM cells treated with Cu/DPA in this study. To confirm the role of TGF-β/Smad signaling in GBM cells, we used CuSO4 and TGF-β to regulate wound healing and the levels of EMT markers, such as N-cadherin, E-cadherin, and vimentin.

Q4. TGF-b signaling can regulates tumor growth through inhibition of cell proliferation or induce apoptosis. The authors should distinguish the effects on those two aspects in their assays.

à TGF-β has been reported to function as a tumorigenic factor by promoting cancer cell and metastasis, as well as being involved in glioma invasion, high-grade tumors, and poor clinical prognosis, which are associated with the TGF-β expression levels. In addition, we show that the expression of TGFβ isomers was reduced by Cu/DPA in GBM cells. To directly confirm the link between TGFβ and EMT properties, we demonstrated in a wound healing assay the levels of EMT markers treated TGFβ and CuSO4 in GBM cells. Therefore, the effect of TGFβ in this study has been distinguished from growth inhibition and apoptosis.

Q5. It is important to show if DPA alone has any effects on EMT phenotypes in GBM cells.

à In the revised manuscript, we modified the sentence: "Neither DPA or TETA affected tumor cell growth or EMT properties in GBM cells and xenograft models. We demonstrated that DPA suppresses the expression of TGFβ signaling properties in the presence of copper and exhibits concentration-dependent cytotoxicity in GBM cells." (line 379)

Q6. It is known that other cells in the tumor microenvironment can be important source of TGF-b rather than cancer cell themselves. Therefore, the effect of DPA in vivo would be more complicated.

à We do agree with the reviewer’s point that the correlation of TGFβ with Cu/DPA may be complicated. Elucidating the detailed source of TGFβ by Cu with chelators is an interesting research topic. We hope such follow-up studies and preclinical studies will be organized.

Round 2

Reviewer 2 Report

I believe the authors have addressed all my previous concerns. The revised manuscript has been improved based on the changes.